# The Role of Oncostatin M and Its Receptor Complexes in Cardiomyocyte Protection, Regeneration, and Failure

**DOI:** 10.3390/ijms23031811

**Published:** 2022-02-05

**Authors:** Thomas Kubin, Praveen Gajawada, Peter Bramlage, Stefan Hein, Benedikt Berge, Ayse Cetinkaya, Heiko Burger, Markus Schönburg, Wolfgang Schaper, Yeong-Hoon Choi, Manfred Richter

**Affiliations:** 1Department of Cardiac Surgery, Kerckhoff Heart Center, Benekestr. 2-8, 61231 Bad Nauheim, Germany; p.gajawada@kerckhoff-klinik.de (P.G.); s.hein@kerckhoff-klinik.de (S.H.); b.berge@kerckhoff-klinik.de (B.B.); a.cetinkaya@kerckhoff-klinik.de (A.C.); h.burger@kerckhoff-klinik.de (H.B.); m.schoenburg@kerckhoff-klinik.de (M.S.); 2Institute for Pharmacology and Preventive Medicine GmbH, Bahnhofstraße 20, 49661 Cloppenburg, Germany; peter.bramlage@ippmed.de; 3Max-Planck-Institute for Heart and Lung Research, 61231 Bad Nauheim, Germany; wolfgang.schaper@mpi-bn.mpg.de; 4Campus Kerckhoff, Justus-Liebig-University Giessen, 61231 Bad Nauheim, Germany; 5German Center for Cardiovascular Research (DZHK), Partner Site RhineMain, 60549 Frankfurt/Main, Germany

**Keywords:** oncostatin M receptor, leukemia inhibitory factor receptor, gp130, interleukin-6, cardiomyocyte, inflammation, dedifferentiation, remodeling, myocardial infarction, heart failure

## Abstract

Oncostatin M (OSM), a member of the interleukin-6 family, functions as a major mediator of cardiomyocyte remodeling under pathological conditions. Its involvement in a variety of human cardiac diseases such as aortic stenosis, myocardial infarction, myocarditis, cardiac sarcoidosis, and various cardiomyopathies make the OSM receptor (OSMR) signaling cascades a promising therapeutic target. However, the development of pharmacological treatment strategies is highly challenging for many reasons. In mouse models of heart disease, OSM elicits opposing effects via activation of the type II receptor complex (OSMR/gp130). Short-term activation of OSMR/gp130 protects the heart after acute injury, whereas chronic activation promotes the development of heart failure. Furthermore, OSM has the ability to integrate signals from unrelated receptors that enhance fetal remodeling (dedifferentiation) of adult cardiomyocytes. Because OSM strongly stimulates the production and secretion of extracellular proteins, it is likely to exert systemic effects, which in turn, could influence cardiac remodeling. Compared with the mouse, the complexity of OSM signaling is even greater in humans because this cytokine also activates the type I leukemia inhibitory factor receptor complex (LIFR/gp130). In this article, we provide an overview of OSM-induced cardiomyocyte remodeling and discuss the consequences of OSMR/gp130 and LIFR/gp130 activation under acute and chronic conditions.

## 1. Introduction

Heart failure (HF) occurs with an incidence of about 2% in the Western population. Statistically, 25–30% of the diagnosed patients are going to die within one year after their initial hospitalization and 50% survive for only five years [1,2]. HF is caused by a broad spectrum of etiologies such as infections, cardiotoxins, hypertension, infarction, obesity, and diabetes [3]. Morphologically and functionally, HF is characterized by multiple adverse cardiac remodeling processes, often associated with inflammation, leading to a continuous loss of the heart’s contractile capacity and ability to supply organs with sufficient nutrients and oxygen. Unfortunately, the scientific view of HF as a simple hemodynamic and neurohormonal disorder has been slow to change over the past decades toward recognizing HF as a complex disorder. This view may be one reason why current therapeutic strategies, as well as the pharmacological armamentarium, are not satisfactory and should be improved. 

The limitation of a merely hemodynamic and neurohormonal explanation of HF, and growing evidence of the involvement of immuno-modulatory molecules in the development of heart failure, led to an alternative working hypothesis, the “cytokine hypothesis” [4,5]. Cytokines are small secretory proteins that function in autocrine, paracrine, or endocrine ways. They regulate various biological processes like hematopoiesis, inflammation, proliferation, differentiation, and the function of innate and acquired immune systems [6]. Cytokines maintain homeostasis under normal conditions, but uncontrolled release of cytokines often leads to disease conditions. According to cytokine hypothesis, increased cytokine levels contribute to the development and progression of chronic HF. Importantly, the cytokine hypothesis marked the beginning of several clinical trials focusing on one of the most promising targets, at that time, tumor necrosis factor-alpha (TNF-α). However, despite a convincing pathophysiological concept, the results of these studies were disappointing and even led to disease exacerbation or death in some patients [4,7]. Nevertheless, the increasing recognition of cytokines, growth factors, and chemokines not only as potential biomarkers but also as promising pharmacological targets has significantly increased scientific interest in identifying specific disease-relevant released molecules and understanding their role in cardiac regeneration and failure.

Another rising star in the cytokine hypothesis is interleukin-6 (IL-6), which gave the IL-6 family its name. Its profound and far-reaching impact on human pathologies has led to an overwhelming number of IL-6 publications. Interleukin-6 releases listed in PubMed in one year (2021) almost equal the total number of publications of its cardioactive family members oncostatin M (OSM), interleukin-11 (IL-11), leukemia inhibitory factor (LIF), and cardiotrophin-1 (CT-1) since their discovery. Considering this imbalance in the number of publications, it becomes evident that there is a significant lack of knowledge about the other members of the IL-6 family. 

While IL-11, LIF, and CT-1 also exert potent effects on cardiomyocytes, we were particularly interested in oncostatin M, as this cytokine was found to be strongly involved in remodeling of the acutely and chronically damaged myocardium in several animal models [8,9,10,11,12]. Moreover, we observed increased activity/expression of components of the OSM system in cardiomyocytes of various human adult heart diseases such as myocardial infarction, myocarditis, aortic stenosis, cardiac sarcoidosis, and ischemic and dilated cardiomyopathy [8,9,10,11,13,14,15,16,17]. Therefore, a major focus of this review was remodeling cardiomyocytes, while other important aspects of OSM activity such as fibrosis, macrophage infiltration, angiogenesis, microRNA, metabolism, proliferation, markers of dedifferentiation, receptor activation, and signaling cascades were not highlighted. Thorough insights about these aspects were given in other specialized publications [9,12,18,19,20,21,22,23,24,25,26].

Since the IL-6 family is thought to have the highest degree of functional pleiotropy and redundancy in triggering the health- and disease-related responses of all cytokine families [27], the unique effects of OSM-activated gp130 receptor complexes feature less in the scientific focus of heart diseases. Therefore, this review aims to provide current knowledge about the unique and general functions of OSM and its receptor complexes in cardiomyocytes, and highlight the consequences of type I and type II receptor complex activation under acute and chronic stress conditions.

## 2. The Interleukin-6 Family of Cytokines and Its Receptors

Since its first cDNA isolation in 1986 by Hirano et al. [28], the pioneer member IL-6 has been joined by the family members CT-1, ciliary neurotrophic factor (CNTF), cardiotrophin-like cytokine (CLC), LIF, OSM, IL-11, IL27, and neuropoietin (NP) [29]. All of these cytokines induce their signals via the common membrane-bound glycoprotein 130 (gp130), which is considered a criterion for membership of this cytokine class [27]. Gp130 is a ubiquitously expressed membrane-bound receptor found in all organs and is critical for survival [30]. Another cytokine, interleukin-31 (IL-31), has been included in the IL-6 family, though it does not bind to gp130. IL-31 signals via a heterodimer of OSMR and IL-31 receptor α (IL-31RA). As IL-31RA and gp130 are located 24 kb apart on the same chromosome and they share approximately 28% homology in their coding regions, IL-31 has been associated with the IL-6 family [31,32]. 

The classical IL-6 signaling cascade is initiated by the binding of IL-6 to the membrane-bound form of the IL-6 receptor (IL-6R), which then triggers complex formation with gp130 (Figure 1). The IL-6/IL-6R/gp130 trimer promotes dimerization, leading to a heterohexameric complex. Such classic IL-6 signaling via the membrane-bound IL-6R is usually restricted to some immune cells and hepatocytes, whereas in cardiomyocytes, IL-6 trans-signaling might play a major role. Trans-signaling requires binding of IL-6 to a soluble extracellular form of the IL-6R, which can be generated by either cleavage of membrane-bound receptors or by alternative splicing. The formation of such IL-6/IL-6R complexes can then activate signaling in gp130-expressing cells [33,34,35]. Similar to IL-6, IL-11 binds to IL11-R, which in turn, activates homodimerization of two gp130 receptors, but such gp130 homodimerization is not required for signaling of all other cytokines of the IL-6 family, which form a heterodimeric complex such as LIFR/gp130, OSMR/gp130, CNTFR/LIFR/gp130, and WSX-1/gp130 [23,36].

The LIFR/gp130 complex is activated by LIF, CT-1, and OSM; OSMR/gp130 by OSM; CNTFR/LIFR/gp130 by CNTF, CLC, and NP, and WSX-1/gp130 by IL-27 [23,36]. The formation of ligand-receptor complexes is similar for all cytokines of the IL-6 family, with the exception of OSM, in all analyzed species. Initially studied human OSM activates both type I and type II receptor complexes. Although the sequence homology between mouse and rat OSM (rOSM) is higher than that between rat and human OSM, OSM signaling activation is similar in rats and humans [21]. In humans and rats, OSM binds to both type I and type II receptor complexes (Figure 1). In mice, however, it instigates its signals only by binding to the type II complex [21,23]. This unique binding of OSM to the type II complex in mice might be useful to study the therapeutic efficacy of OSMR signaling (Figure 1). To the best of our knowledge, the involvement of IL-6 family members in adult cardiomyocyte remodeling is restricted to IL-6, IL-11, CT-1, LIF, and OSM, as listed in Figure 1, as no or scarce experimental or literary evidence has been found linking the heart to CNTF, CLC, NP, and IL-27 signaling. Therefore, only those members of the IL-6 family that have been clearly shown to be involved in cardiomyocyte remodeling are discussed in this review.

## 3. The Role of IL-6 in Cardiac Regeneration and Failure—Cause or Consequence?

Most members of the IL-6 family activate multiple signaling cascades such as the Jak/STAT, MEK/Erk, Erk5, SAPK/JNK, p38, and PI3 kinase/Akt pathways. Such redundant cascade activation might be explained by the commonly shared coreceptor gp130, which is likely expressed on all cells of the human body. Technically, tissue-specific effects of each family member are determined and evaluated by the cellular expression or silencing of their respective specific receptors. Being at the center of the IL-6 class, gp130 is vital for cardiac development, regulation of cardiac physiology, and remodeling in response to pathophysiological stimuli [37].

IL-6 has emerged as a promising biomarker and potential target in the development of heart disease. However, despite a number of experimental results supporting the idea of IL-6 as an important player in cardiac protection, repair, and failure, this assumption is far from being clear. Activations of the Jak/STAT pathway are often considered part of the IL-6 cascade [38], neglecting the involvement of other IL-6 family members in the development of heart failure. This causal view is supported by increased circulating levels of interleukin-6 in patients with heart disease, as well as increased cardiac expression of IL-6 in animal models of myocardial infarction, cardiac hypertrophy, and dilated cardiomyopathy [39,40,41,42]. Furthermore, IL-6 expression was observed in cardiomyocytes of the border zone in the infarcted area [43]. However, there are also indications that IL-6 might function as a biomarker of cardiac injury rather than playing a critical role in cardiac protection or the development of heart failure. Compared with other members of the IL-6 class, as well as further cardioactive growth factors such as FGFs and IGFs, interleukin-6 is a relatively weak trigger of adult cardiomyocyte remodeling [8,10,44,45,46]. In cultures of adult cardiomyocytes, strong expression and secretion of IL-6 have been shown after interleukin-1β treatment [47] and were observed after IL-1α stimulation (Figure 2A). Moreover, OSM proved to be a potent stimulator of IL-6 secretion, whereas LIF, CT-1, TNF-α, and TGF-β induced little or no change in cultured adult cardiomyocytes after 36 h (Figure 2A). Similarly, OSM also induced a strong release of IL-6 from astrocytes [48], fibroblasts [49], cancer cells [50], cerebral endothelial cells [51], and smooth muscle cells [52]. Although IL-6 has been associated with the infarct size and cardiac function of STEMI patients and in a large cohort of heart failure patients, such studies associating IL-6 could not conclude the causality [53,54].

Animals overexpressing either the human IL-6 or the human IL-6R did not show any myocardial abnormalities, although cross-species binding capabilities have to be considered [55]. Mice lacking IL-6 did not show adverse effects as anticipated on induction of myocardial infarction, suggesting the compensatory activation of other IL-6 family cytokines [56]. Double-transgenic mice overexpressing human IL-6 and soluble IL-6R showed no myocardial phenotype at the age of two months but developed cardiac hypertrophy at five months of age [55], suggesting that cardiac remodeling requires chronic exposure to IL-6/IL-6R signaling. Thus, the direct involvement of interleukin-6 in acute and chronic heart disease is less clear than assumed and requires reevaluation of its function as a causative agent and/or biomarker, and the other cardioactive family members of the interleukin-6 class need to be considered more as players in cardiac remodeling.

## 4. The LIF/LIFR/gp130 Axis in Cardiac Regeneration and Failure

Leukemia inhibitory factor (LIF) was cloned and given its name in 1987 as a suppressor of the proliferation of myeloid leukemia cells in mice [57]. Four years later, its receptor was cloned by screening a cDNA library from the human placenta [58]. It was, therefore, not surprising to find that LIF is crucial from the very early stages of life, as it promotes self-renewal and differentiation of embryonic stem cells as well as blastocyst implantation [59,60]. Subsequently, LIF has been shown to possess great pleiotropic potency by playing regulatory, inhibitory, and inducive roles in various biological processes [61,62]. LIF signaling is initiated by binding to the membrane-bound LIFR, which then dimerizes with gp130, leading to the activation of the Jak-STAT, MAPK, and PI3-K/Akt pathways [63,64]. 

LIF is a very potent remodeling factor for adult cardiomyocytes [8] (Figure 2D) and cardiac functions of LIF have been observed in several experimental models. LIF was shown to be cardioprotective under ischemic conditions by coping with oxidative stress and preventing cell death [65,66]. Preconditioning of a rabbit heart with LIF was shown to be cardioprotective by MnSOD mediated the scavenging of reactive oxygen species (ROS) [67]. In a murine model of myocardial infarction, intramuscular injection of LIF cDNA enhanced cardiomyocyte survival and cardiac regeneration [68]. Targeted overexpression of LIF in rats on myocardial infarction exerted protection by preserving the myocardium and reducing fibrosis [69]. Moreover, hemodynamic overload alone was sufficient to increase the expression of LIF in the heart, which in turn, protected the heart by blocking apoptosis and stimulating cardiac hypertrophy [70]. Increased LIF expression was also observed in a canine model of congestive heart failure [71] and patients with end-stage dilated cardiomyopathy [72].

## 5. The OSM/OSMR/gp130 Axis and Early Cardiovascular Studies

Similar to LIF, OSM also received its name due to its ability to inhibit the proliferation of human tumor cell lines. Long before its cDNA was cloned [73], human OSM was isolated from conditioned media of PMA-treated lymphoma cells and supernatants of activated T-lymphocytes [74]. Unlike LIFR, the cDNA of the OSM receptor (OSMR) was cloned almost a decade after OSM was isolated. The reason for the delay in the discovery of the unique OSMR may be that human OSM also activates the LIFR/gp130 complex [75]. On cDNA cloning of the OSM receptor, it was found that human OSM could bind and activate both type I and type II complexes [76]. Not surprisingly, the overlapping binding capabilities of human OSM are consistent with its genetic location. OSM and LIF genes are located only 10 kbp apart in the same transcriptional direction on human chromosome 22q12. Their proximity on the chromosome and the similarity of their exon/intron structures indicate that both cytokines originated by gene duplication [77]. Likewise, their receptors are also located in close proximity on the same chromosome [78] and they too might have arisen from ancient gene duplication [79]. Such local duplication led to a distinct ligand-specific receptor chain for OSMR [80]. Similar to LIF, OSM is also involved in several diseases of organs and tissues, such as the central nervous system, liver, heart, connective tissue, bones, lung, the vascular system, and different types of cancer [81]. OSM is linked to diverse metabolic processes such as bone metabolism, cartilage catabolism, LDLR expression in hepatocytes, and also maintenance of homeostasis in the central nervous system [81]. 

Accumulating evidence suggests that OSM is involved in organ and tissue regeneration as well as fighting infectious diseases. In concert with IL-6, OSM contributes to the survival and proliferation of hepatocytes after acute liver injury through the expression of tissue inhibitor of metalloproteinases-1 (TIMP-1) [82]. Similar to LIF and IL-6, OSM was also present at high concentrations in the bronchoalveolar lavage from patients with acute lung injury. Interestingly, OSM levels were significantly higher in patients with pneumonia than in patients with acute liver injury [83]. Moreover, it is intriguing to see elevated levels of OSM, similar to IL-6 and TNFα, in the plasma of COVID-19 patients showing severe symptoms and those who were admitted to the intensive critical unit (ICU). Interestingly, changes in OSM mRNA were not observed in peripheral blood mononuclear cells, suggesting a tissue origin for OSM [84]. The importance of OSM during COVID-19 development can be inferred from the cluster analysis of selected markers, which showed both IL-6 and OSM as the markers that differentiated ICU and non-ICU patients [85]. Astonishingly, 7–8 months after being infected with SARS-CoV-2, both asymptomatic and symptomatic patients showed elevated levels of OSM in serum [86]. Based on its strong ability to induce IL-6 in various cell types, OSM could play a larger role in SARS-CoV-2 infection than anticipated. Likewise, exponential high levels of circulating OSM were measured in patients who developed infections after implantation of left ventricular assist devices [87] and in patients suffering from septic shock [88].

Its ability to enhance the expression of adhesion molecules on endothelial cells, induce angiogenesis, and regulate smooth muscle cell growth suggests a proatherosclerotic function for OSM [89,90,91]. Moreover, OSM has also been detected in atherosclerotic lesions of the human aorta and ApoE-deficient mice [92]. However, a recent study showed that the development of atherosclerosis was reduced on chronic administration of OSM, and high OSM levels coincided with an increased post-incident coronary heart disease survival probability in the AGES-Reykjavik study [93], indicating the complexity of correlation studies. 

Although the OSMR is expressed by cardiomyocytes, its cardiac role is poorly understood. Studies performed in cardiomyocyte cultures showed that OSM regulates the expression of modulators of extracellular matrix degradation such as plasminogen activator inhibitor 1 and TIMP-1 [94,95]. During the inflammatory healing phase, OSM appears to counteract excessive matrix degradation by proteases, thereby influencing cardiac repair and remodeling. This beneficial property of OSM on cardiomyocytes is reinforced by evidence that it induces the proangiogenic vascular endothelial growth factor (VEGF) [96]. The angiogenic potential of OSM has been further ascertained in vivo in a rabbit corneal model and a murine model of myocardial ischemia [25,97]. In addition to inducing angiogenesis, OSM was shown to secrete a stem cell quiescence factor, stromal cell-derived factor 1α (SDF1), in adult cardiomyocytes [98]. Coincidently, recruitment by the VEGF/SDF1 nexus of progenitor cells was unveiled during myocardial repair [99].

A link between OSM and fetal remodeling in the stressed myocardium is already supported by early studies. OSM expression was detected in the ischemic border zone of the reperfused canine myocardium [43]. In culture, adult cardiomyocytes treated with OSM showed increased viability, decreased contractility, sarcomeric loss, and striking dedifferentiation [100,101], suggesting life-saving functions in hypoperfused remodeling hearts. In cardiomyocytes, although OSM activates the MEK/Erk, Jak/STAT, PI3 kinase-Akt, P38, Erk5, and SAPK/JNK pathways, the inhibition of the MEK/Erk pathway alone was sufficient to abolish OSM-induced cardiomyocyte dedifferentiation [8]. Loss of contractile myofilaments and dedifferentiation were observed in the hibernating myocardium, in atrial myocytes during fibrillation, in hearts supported by assist devices, and in the border zone of infarcted regions [102,103,104,105,106,107,108,109]. This fetal remodeling, including loss of contractile filaments, is thought to enable cardiomyocytes to tolerate adverse cardiac conditions and survive episodes of hypoperfusion, thus preventing irreversible damage at an early stage. However, prolonged periods of hypoperfusion increase cardiomyocyte degeneration and impair recovery of function after the restoration of blood flow [107,109].

## 6. The Complexity of Determining the Function of OSM in Human Heart Diseases

While LIF is considered the most pleiotropic member of the IL-6 family [57], the ability of human OSM to activate both type I and type II receptors suggests an even greater difficulty in determining biological and pathologic functions (Figure 1). Moreover, OSMR signaling is likely to be even more complex, as it has been demonstrated that not only does the use of other activated receptors lead to the amplification of OSM signals but also that the OSMR is directly involved in unrelated receptor signaling cascades. In OSM-treated cardiomyocyte cultures, activation of the interleukin-4 receptor (IL-4R) strongly enhanced the release of FGF23, whereas the IL-4 receptor ligands IL-4 and IL-13 themselves did not trigger secretion [110]. FGF23 was initially identified as an endocrine hormone affecting mineral metabolism and decreasing phosphate reabsorption in the kidneys, but recent studies demonstrated its association with cardiac diseases [11,17]. In human fibrosarcoma cells, treatment with OSM resulted in rapid and pronounced phosphorylation of the signal-transducing subunit of the interferon-γ receptor [111].

Moreover, in glioblastoma cells, OSMR serves as an essential co-receptor for EGFRvIII—a truncated, active form of epidermal growth factor receptor—and its silencing or pharmacological inhibition of the EGFRvIII-OSMR interaction suppressed Stat3 activation, cell proliferation, and tumor growth [112]. Thus, it is clear that despite sophisticated transgene and knockout technologies, the analysis of OSM-activated receptor signaling cascades in vivo poses the problem of enormous complexity, which is especially true for understanding human pathologies. Therefore, it is reasonable to use simplified in vitro assay systems as a first step to decipher OSM receptor-dependent properties. 

Cultures of primary adult rat cardiomyocytes provide a simplified and useful model to gain essential insights into the biological and pathological functions of OSM in the adult heart. In contrast to cell lines such as HL-1 or induced pluripotent stem cells, primary isolates transfer their genetic, epigenetic, and proteomic status to the culture dish with few experimental artefacts, and therefore, offer high reliability in identifying disease-relevant targets in experimental studies [3,8,100]. Furthermore, species-specific and cross-species binding capabilities of the human, rat, and murine OSMs make the rat an ideal experimental toolbox to study individual as well as simultaneously activated receptor complexes (Figure 1) and help to decipher the underlying signaling cascades in detail. In rats, human OSM activates the LIFR complex but not the OSMR complex, and murine OSM binds only the OSMR complex, whereas rOSM activates both type I and type II receptor complexes [21,23]. Because mOSM activates only the OSMR complex in mice, this rodent does not offer the same potential as the rat for studying human diseases, and the above-mentioned cross-species binding capabilities of OSM must be considered when interpreting early studies in mice. In addition to the potential for selective activation of OSM receptor complexes, adult rat cardiomyocyte cultures are well-suited to siRNA-based transcriptome and proteome studies that allow for detailed analysis of signal transduction cascades and the effects of a single knockdown on cardiomyocytes. For example, B-Raf, a signaling component of the Ras/Raf/MEK/Erk cascade, has been identified as a master regulator of ischemic resistance in adult cardiomyocytes [8,113]. Moreover, these cultures use mechanisms similar to cardiomyocytes in the damaged myocardium to re-establish cell-cell contacts and eventually form a synchronously contracting tissue-like layer with well-established sarcomeres [3,10,15,24,46,100,114,115]. Below, we will outline the remodeling patterns of adult rat cardiomyocytes and show how insights into OSM gained from primary cell cultures can be applied to understand human heart diseases.

## 7. Effects of Type I and II Receptor Activation on Cultured Cardiomyocytes

The most commonly used growth stimulant and maintenance medium for cardiomyocytes is fetal calf serum (FCS), which is usually used at concentrations between 5% and 20%. FCS induces hypertrophic remodeling of cultured adult cardiomyocytes characterized by cell surface enlargement, protein accumulation, and restoration of cell-cell contacts by in situ spreading, resulting in a cross-striated confluent layer of contracting cardiomyocytes with time [46,100,113]. Importantly, serum from human patients with coronary artery disease and dilated cardiomyopathy elicits similar and even more rapid effects, indicating a comparable composition of cardioactive growth factors [100,113,116]. In contrast, stimulation with secreted factors (morphogens) from cardiac microvascular endothelial cells of patients with dilated cardiomyopathy results in fetal remodeling characterized by severe cardiomyocyte elongation and rapid restoration of cardiomyocyte-cardiomyocyte contacts, atrophy, and the activation of the fetal gene program [116]. The ability of cardiomyocytes to restore cell-cell contacts is an important issue in the remodeling myocardium because myocytes are not only connected in a row at the cell ends via intercalated discs but also form a distinct three-dimensional structure by laterally arranged intercalated discs (Figure 2B,C). While adult cardiomyocytes exert tremendous force during contraction, their susceptibility to deformation and twisting makes them highly vulnerable, which is reflected in the isolation procedure [3,24]. Adult cardiomyocytes are obtained and isolated by digestion of the extracellular matrix on a Langendorff perfusion system, whereas neonatal myocytes are directly minced with scalpels in digestion solution for successful isolation [117,118]. Any steps that deviate from a strict isolation protocol will result in a reduction in the viability and quality of adult cardiomyocytes.

In ischemic cultures, fetal remodeling results in the protection of adult cardiomyocytes in a way that was little observed after hypertrophic remodeling [100,113,116]. These observations lead to three conclusions that are integrated into an overall scheme of cardiac adaptation, regeneration, and failure, as discussed later: First, the growth factor/cytokine compositions of morphogens and sera differ substantially. Second, locally produced morphogens and circulating factors (serum) exert different functions in the diseased heart. Third, the accessibility to these factors in the heart must be spatially and temporarily regulated. In searching for factors responsible for the reexpression of fetal genes, the reestablishment of cardiomyocyte-cardiomyocyte contacts, and the induction of survival pathways, screening of secreted factors from cultured rat, porcine, and human microvascular endothelial cells was performed, which finally led to the identification of LIF and OSM as major remodeling factors [8,41,100,101,113]. 

In culture (Figure 2D), one of the most obvious effects of the interleukin-6 cytokines is cell lengthening, which is most prominent after OSM stimulation and can be readily detected by bright field microscopy [8]. When rat adult cardiomyocytes are initially plated in 2% FCS and then kept serum- and insulin-free for a prolonged period of 5–7 days, cardiomyocytes increase relatively little in their surface area and width, but their length increases several-fold (Figure 2D). Lateral thinning and length extensions are more pronounced with rOSM stimulation than with mOSM. The activation of both type I and type II receptor complexes in rat cardiomyocytes might be the reason for such drastic morphological changes. The development of multiple extensions from the area of the intercalated discs reestablishes cell contacts with both adjacent and distant cardiomyocytes and leads to the formation of a distinct two-dimensional network in culture. Lateral cell-cell contacts are not particularly pronounced in mOSM and rOSM stimulated cultures, but limited cell spreading might be observed when the LIFR/gp130 complex is activated. The increasing extent of elongation induced by IL-6 family members can be observed in the following order: IL-6 < IL-11/CT-1/LIF < mOSM < rOSM. It has to be emphasized that in our research, the usage of prescreened sera [44] was necessary to determine the effects of IL-6 on adult cardiomyocytes because IL-6R is hardly expressed on cardiomyocytes, suggesting trans-signaling effects [33,37], or alternatively, that serum induces IL-6R expression in cardiomyocytes. In the continuous presence of serum, there is also a marked increase in surface area due to the restoration of lateral cell contacts in cultures stimulated with IL-6 family members (not shown). In contrast to IL-6 cytokines, FGF-2 induces an increase in the surface area, leading to lateral cellular contacts, and also some extensions might be formed (Figure 2D). The presence of serum enhances FGF2 effects. FGF2 is an effective trigger of hypertrophic remodeling as this growth factor induces the formation of new sarcomeres in cultures of adult cardiomyocytes [114,119]. Another feature of OSM—observed to a much lesser extent in LIF-treated cultures—is dedifferentiation and sarcomeric loss in cultured adult cardiomyocytes with a concomitant decrease in contractility (summarized in Table 1). In the presence of mOSM, fetal remodeling can be detected by the reexpression of non-muscle α-actinins, smooth muscle α-actin, destrin, moesin, and the stem cell markers Runx-1, Dab2, and c-Kit (Table 1). There are certainly many more dedifferentiation markers to be discovered as the effects of mOSM on cardiomyocytes are quite dramatic and the currently identified remodeling markers are probably only the “tip of the iceberg”. Despite the fact that dedifferentiating cardiomyocytes show similar patterns of fetal remodeling in humans, mice, and rats, there are certain differences that should be noted. One of the most highly and relatively rapidly reexpressed genes in rodents is smooth muscle α-actin, which is detectable but rarely expressed in human dedifferentiating cardiomyocytes [11]. In contrast, non-muscle α-actinin-1 is relatively late to be reexpressed in rat cardiomyocyte cultures, but unlike smooth muscle α-actin, appears to be an important dedifferentiation marker in human heart disease [8,13,14,113,116].

The continuous loss of sarcomeric structure, as evidenced by the disappearance of the cross-striated pattern, is accompanied by the downregulation of sarcomere proteins such as myomesin, titin, and cardiac myosin heavy chains at both the protein and mRNA levels (Table 1). The loss of contractile capabilities and downregulation of the oxygen carrier myoglobin could contribute to the dramatically decreased beating activity of adult cardiomyocytes. Sarcomeric degeneration has been shown to be dependent on increased expression and activity of the matrix metalloproteinase-2 in cardiomyocytes [120]. The mechanism by which OSM downregulates beating activity appears to be distinct from TGF-β as this growth factor inhibits remodeling and preserves the cardiomyocyte structure in cultures treated with serum from patients with coronary artery disease [115]. Interestingly, strong expression of TGF-β has been observed in hibernating myocardium, and its functional properties may also contribute to the status of hibernation [115]. 

The ability to activate two distinct receptor complexes, as well as its ability to convert signals from other cardioactive receptors into fetal remodeling processes, makes OSM a dominant regulatory cytokine in the stressed myocardium. For example, FGF2 is a potent hypertrophic growth factor that is evident in the surface area enlargement of cardiomyocytes and affects their myofibrillar growth and beating activity [45,46,114], but in the presence of OSM, fetal remodeling [12] is accelerated and cardiomyocytes become quiescent. Similar properties of OSM apply to IGF-1-treated cardiomyocytes [10]. Furthermore, integration of IL-4 receptor signals—which cause only minor changes in FGF23 secretion—strongly potentiate FGF23 release [110]. Another important feature of OSM is its ability to affect the myocardial environment by inducing the strong production and release of potent extracellular proteins (Table 1) such as TIMP-1, interleukin-6, and the regenerating islet-derived proteins (Reg1 and Reg3 chemokines) [9,11]. The release of FGF23 from the stressed myocardium into the circulation suggests that OSM also exerts systemic effects through the release of potent molecules from cardiomyocytes, but conclusive evidence is still lacking [11,16]. Overall, these observations clearly indicate important functions of OSM in both the acutely and chronically injured myocardium, as are described below.

## 8. The OSM/OSMR/gp130 Axis Exerts Protection and Improves Regeneration after Acute Myocardial Infarction

The release of OSM infiltrating macrophages and increased expression of the OSMR are processes that are involved in both protection after acute myocardial damage and the development of chronic heart failure as discussed in the next section. Similar to cultured adult rat cardiomyocytes treated with OSM, four main effects related to increased expression and activation of the OSMR complex can be observed in cardiomyocytes of the infarcted mouse myocardium. First, reorganization of the cardiomyocyte structure results in fetal remodeling, as evidenced by reexpression of fetal genes such as non-muscle α-actinins, smooth-muscle α-actin, destrin, moesin, and stem cell markers Dab2 and Runx1 in the myocardium [8,10,11,13,14,15,121,122,123]. The reexpression of B-Raf [8,113], which is hardly expressed in normal cardiomyocytes, indicates that fetal reprogramming is not only limited to altering cellular morphology but also affects the quality of OSM receptor signaling cascades such as the Ras/Raf/MEK/Erk (MAPK) pathway in the stressed myocardium. B-Raf has been shown to be an essential signaling transducer in the ischemic survival machinery [8,113]. Second, the resolution of the intercalated discs (Figure 3A), as well as the three-dimensional cellular contacts, is accompanied by cardiomyocyte elongation (Figure 3B). Third, the loss of the sarcomeric structure and downregulation of the O_2_-carrier myoglobin reflects the decreased activity of the oxygen consumption machinery and metabolic remodeling [10,12,15,24]. Fourth, the release of various potent molecules such as FGF23, TIMP-1, ANP, BNP, and interleukin-6, as well as the regeneration of islet-derived proteins, not only influences the cardiac environment but might also mediate systemic effects. These four properties appear to be essential for protective effects in the infarcted area.

Further evidence of a protective key role is provided by studies using recombinant OSM or OSMR knockout mice. Therapeutic application of recombinant mOSM markedly improved cardiac performance and dramatically increased survival after infarction, whereas strains with a deletion of OSMR showed a reduced ejection fraction and high mortality [8,9,12]. Similar observations in OSM-treated and OSMR knockout animals were made by other groups, which showed improved cardiac performance, reduced fibrosis, and angiogenesis in the infarct zone involving VEGF, FGF-2, and TIMP-1 [12,25,124]. In addition, OSM likely amplifies its signals by upregulating its own receptor and enhances incoming FGF signals by strongly inducing FGF receptor 1 expression in cardiomyocytes [12]. In a recent publication, OSM released by hypoxic macrophages was shown to directly inhibit the TGF-β-mediated activation of cardiac fibroblasts and reduce the development of cardiac fibrosis in a mouse model of trans-aortic constriction [18]. In a mouse model of aortic stenosis, circulating FGF23 levels increased sharply, whereas deletion of the OSM receptor resulted in nearly the same level of this growth factor as in sham-operated control animals [16]. Similarly, elevations of serum FGF23 were associated with OSM-expressing infiltrates in the myocardium and with FGF23-positive cardiomyocytes in patients with aortic valve defects [11,16].

There is increasing evidence that OSM acts as an acute response cytokine. Stimulation of different cell lines with a subset of cytokines from myeloid and lymphoid cell lines increases OSM expression within 30 min [125]. The production of acute-phase response proteins is impaired in OSMR knockout mice after partial hepatectomy as well as after CCL_4_ hepatotoxin treatment [82]. In a canine model of ischemia/reperfusion, expression of OSM was observed within the first hours of reperfusion in infiltrating and endothelial cells of the ischemic myocardium [43]. Furthermore, OSM degranulation and production have been described for alveolar neutrophils from patients with acute lung injury [126] and this cytokine is also regarded to play a dominant role in the renal acute-phase response [83]. The ability of LIF to induce acute-phase response proteins reinforces the view that OSM, as an activator of type I and II receptor complexes, functions as an early regulator of stress defense in human diseases [127]. In the infarcted myocardium, OSM is released within a few hours by invading neutrophilic granulocytes. Initially, it might be released by degranulation of preformed stocks and then by de novo synthesis [126]. Later, neutrophils are replaced by OSM-secreting macrophages [9]. Transcriptome analysis of myeloid cells isolated from the infarction zone showed a high expression of OSM after four days of infarction, which was higher than all other IL-6 family members [9].

Upregulation and activation of the OSMR in cardiomyocytes will not only channel resources to overcome the ischemic environment by MAPK-activated fetal remodeling but also provide means to orchestrate the infiltration of OSM-releasing macrophages through the secretion of Reg chemokines [9,13]. The ability of OSM-releasing macrophage infiltration to ensure ischemic resistance of dedifferentiating cardiomyocytes was demonstrated in a mouse strain with cardiac overexpression of monocyte chemotactic protein-1 [128,129]. Conversely, fetal remodeling of cardiomyocytes and macrophage infiltration of the infarct area were strongly downregulated in OSMR and Reg3β knockout strains, which was associated with markedly reduced survival [8,9]. Given these properties, it would not be surprising if OSM accounts for at least part of the regenerative capacity of stem-cell therapies and acts as an important immunomodulator in the infarct area. Evidence for this can be derived from several experimental approaches. In a mouse model of myocardial infarction, the application of OSM-secreting skeletal myoblasts, as well as embryonic stem cell-derived cardiomyocytes, significantly improved the cardiac function [44]. Such beneficial effects are nowadays attributed to paracrine signaling cascades rather than to the formation of newly formed contractile cardiac tissue by transplanted cells [44,130,131]. Zymosan is known to enhance the immune responses through activation of the macrophage toll-like receptor-2 (TLR2). Injection of Zymosan into the infarcted area of mice increased the number of activated CD68^+^ macrophages and significantly improved ventricular performance [131]. Furthermore, chitin, which is regarded to be a relevant TLR2-stimulatory component of zymosan, induced strong expression of OSM in human whole-blood samples [132]. Because cyclosporine A is also a potent direct inhibitor of cardiomyocyte remodeling in OSM, LIF, or morphogen-treated cultures (unpublished), the view of cyclosporine A as an inhibitor of acute cardiac infarct regeneration by merely suppressing immune cell activation should be interpreted with caution. All these data suggest that OSM is an important pivot of cardiac protection and regeneration by orchestrating immune cell activation, infiltration, and fetal remodeling of cardiomyocytes. Clearly, the cycle of OSM-releasing infiltrating cells and chemokine-releasing cardiomyocytes in the injured area must be interrupted to prevent chronic inflammation and deleterious remodeling, as discussed below. 

## 9. Chronic Activation of the OSM/OSMR/gp130 Axis Contributes to the Development of Heart Failure

Detrimental effects of chronic infiltration have been demonstrated in a mouse strain with cardiac-restricted overexpression of the monocyte chemotactic protein-1 (MCP-1) [128]. This adverse remodeling appears to be the consequence of a reduction in contractile power caused by chronic Ras/Raf/MEK/Erk activation, cardiomyocyte dedifferentiation, and sarcomeric loss. These animals develop postnatal myocarditis, which manifests over time dilated cardiomyopathy and culminates in end-stage heart failure. By four months of age, hearts are characterized by massive infiltration of OSM-releasing macrophages and excessive fetal remodeling of cardiomyocytes [8,10,11,128]. Remodeling of the heart in this MCP-1 strain mimics remodeling of OSM-treated cell cultures, with six features. First, by activation of the OSMR/gp130 cascade as evidenced by phosphorylation of gp130 and Erk1/2, as well as by strong upregulation of OSMR and B-Raf; second, by reexpression of non-muscle α-actinin-1, smooth-muscle α-actin, moesin, and destrin; third, by the resolution of the intercalated discs; fourth, by the reexpression of the stem cell markers Runx1 and Dab2; fifth, by the downregulation of the oxygen consumption machinery, and sixth, by strong upregulation of secreted proteins such as atrial natriuretic peptide (ANP) and FGF23. The reexpression of non-muscle α-actin, ANP, and phosphorylation of gp130 and ERK1/2 suggest but are not necessarily indicative of exclusive OSM activity, whereas the reexpression of stem cell markers and FGF23 are likely specific for OSM since no other factors regulate these genes in the heart. The adverse effects of chronic macrophage infiltration in these transgenes were partly reversed by pharmacological treatment with an OSMR antibody or by the genetic deletion of one or both OSMR alleles. Inhibition of the OSMR signaling cascade not only significantly reduced the extent of deleterious cardiac remodeling but also increased the life expectancy of these chronically infiltrated transgenes [8,10]. Similar to the MCP-1 transgenes, infiltrates of OSM-releasing macrophages and increased OSMR expression in cardiomyocytes can be observed in patients with an inflammatory background such as myocarditis (Figure 4A) or cardiac sarcoidosis [10,13]. Disruption of the typical architecture of the intercalated discs (Figure 3D) and formation of cardiomyocyte extensions are also striking and appear to be independent of the underlying inflammatory disease (Figure 4B). FGF23, a marker of OSM activity, can be observed as single FGF23-positive and clusters of positive cardiomyocytes (Figure 4C). The development of chronic inflammation might not only be linked to increased levels of inflammatory mediators but also relate to insufficient or decreased levels of anti-inflammatory molecules. Mice with a deletion of the anti-inflammatory TGF-β showed no gross developmental abnormalities, but these animals developed myocarditis and died within the first weeks after birth due to an uncontrolled inflammatory response [133,134]. Uncontrolled inflammatory processes are well-known in humans with a deactivating mutation of the interleukin-1 receptor antagonist (IL-1Ra; DIRA), and daily treatment with a recombinant form of IL-1ra (Anakinra) reverses inflammation and reduces lethality [135]. In the myocardium of transplanted patients with cardiac sarcoidosis, IL-1ra is strongly expressed in macrophages of the granuloma, reflecting reduced activation of certain stress and inflammatory pathways [13]. The successful use of recombinant IL-1 blockers in experimental models suggests that these antagonists represent a promising therapeutic strategy for the treatment of patients with heart diseases [135,136]. 

Similar to animal studies, sustained activation of the OSMR/gp130 cascade correlates with macrophage infiltration and the development of heart failure in adult patients with various chronic cardiac diseases. Increased accumulation of Reg3A (the human ortholog of Reg3β in mice and rats) and Reg3γ was observed in dedifferentiating cardiomyocytes surrounding the granuloma, correlated with increased OSM levels [13]. Elevated levels of Reg3A were also found in patients with ischemic heart disease [9]. Increased expression of Reg1, which is strongly induced by OSM in cultured cardiomyocytes, was observed in patients who died of myocardial infarction and is also expressed in cardiomyocytes adjacent to the granuloma in cardiac sarcoidosis patients [9,137].

Activation of the Jak/STAT pathway has been described in various human cardiac diseases, but it should be kept in mind that activation of this signaling cascade cannot be attributed entirely to a particular member of the IL-6 family. To distinguish an OSMR-mediated effect from other IL-6 receptor complexes, one must look for more specific effects of this cytokine. ANP and BNP are regarded as biomarkers for cardiac remodeling induced by various cardioactive agents including FGF2, but only the increasingly recognized cardiac remodeling marker FGF-23 appears to be specific to cardiac OSM activity [11,17,138,139,140]. Other strong indicators of fetal remodeling are non-muscle α-actinins (Actn1 and Actn4), which are expressed during the early postnatal weeks in the normal human myocardium but are barely detectable in adult cardiomyocytes [116]. Reexpression of Actn1 in cardiomyocytes is also a strong indicator of OSM activity. Actn1 was detected in almost 10% of all cardiomyocytes in the majority of patients with aortic stenosis and dilated cardiomyopathy, who had a severely reduced ejection fraction [8,10,14]. Moreover, reexpression of this non-muscle actinin and cardiac accumulation of Reg3A, Reg3, and OSM in patients with cardiac sarcoidosis correlate well with macrophage recruitment and granuloma formation [13]. In that context, it should be noted that other members of the IL-6 family also induce Reg3β expression in cardiomyocytes, and it is, therefore, very likely that OSM induces a much greater release in humans than in mouse cardiomyocytes by activation of type I and type II receptor complexes [9]. Further evidence of OSM participation in human cardiac pathology includes the reexpression of Runx1 and moesin. Moesin belongs to the cytoskeletal linker proteins (ERM, Ezrin, Radixin, Moesin) and is usually poorly expressed in normal cardiomyocytes, but together with radixin, marks quite dramatic structural changes of the intercalated discs in patients with end-stage heart failure [3,15,116]. However, it should be kept in mind that reexpression and relocalization of moesin can also be triggered by other cardioactive interleukin-6 family members and hypertrophic growth factors such as FGF2 and IGF1. Below, we summarize the functions of OSM and coregulators in an overall hypothetical scheme of cardiac repair and failure.

## 10. Translating Knowledge about OSM into a Hypothetical Overall Model of Cardiac Repair and Failure

Because OSMR activation has been observed in patients with both inflammatory and non-inflammatory diseases, it is unlikely that OSM release in the myocardium necessarily causes myocardial inflammation per se. One mechanism to prevent uncontrolled inflammatory responses could be the antagonistic upregulation of anti-inflammatory mediators such as TGF-β [141] and the IL-1 receptor antagonist [13,142]. The temporal expression and localization of anti-inflammatory mediators might differ between cardiac diseases and could be due to disease-related defense mechanisms, as seen for IL-1ra in patients with ischemic cardiomyopathy and cardiac sarcoidosis (Figure 5). In patients with ischemic cardiomyopathy, IL-1ra is mainly present in cardiomyocytes (Figure 5A), whereas accumulation of this antagonist is essentially observed in macrophages of patients with cardiac sarcoidosis (Figure 5B).

Under conditions of increased stress, spatially and temporarily limited activation of the OSMR cascade supports the reorganization of the intercalated discs (ID) to avoid deleterious distortion and provide the basis for an enhanced myofibrillogenesis. As a result of hypertrophic growth, cardiomyocytes will increase in length and width, with a marked accumulation of sarcomeres (Figure 6). At an early stage, some degree of hypertrophy is reversible, for example, after valve replacement or pregnancy, since cardiomyocyte contacts are essentially preserved and functional. The increase in growth-promoting substances such as fibroblast growth (FGFs) and insulin-like growth factors (IGFs), combined with a concomitant decrease in OSM receptor signaling cascades, supports the development of hypertrophic cardiomyocytes. However, when dying or degenerating cardiomyocytes disrupt intercellular communication, mechanical transmission of force is impaired and life-sustaining fetal remodeling of cardiomyocytes must occur (Figure 6). This is especially evident after acute myocardial infarction. Invading neutrophil granulocytes activate OSMR and LIFR signaling cascades in cardiomyocytes and initiate repair processes but it is also likely that other members of the IL-6 family participate. Subsequent infiltration and regeneration of the damaged myocardium are controlled by several classes of chemokines, such as the regenerating islet-derived proteins (Regs), monocyte chemoattractant proteins (MCPs), and Interleukin-7 (IL-7), whose receptor is significantly expressed on most cardiac macrophages of the infiltrated area [143]. Concurrent with increased expression and activation of the OSMR, cardiomyocytes in the ischemic area undergo fetal remodeling and the oxygen consumption machinery is downregulated. The release of ANP, BNP, and FGF23 into the circulation is accompanied by the reexpression of non-muscle actinins, FGF23, and moesin, as well as by relocalization of the ERM proteins in remodeling cardiomyocytes. The natriuretic peptide ANP and the N-terminal precursor NT-proBNP are commonly used remodeling markers in clinical settings [144]. In addition, increases in circulating FGF23 have been shown to be associated with left ventricular dysfunction and ventricular hypertrophy [138]. Similar to OSM-treated cell cultures, cardiomyocytes in the myocardium lose the typical architecture of the intercalated disc and the formation of extensions becomes visible (Figure 6). However, it is important to note that changes in the architecture of the intercalated disc are not necessarily limited to a single heart disease but may also be observed in cardiac diseases triggered by various causes such as in patients with myocarditis, cardiac sarcoidosis, and ischemic and dilated cardiomyopathy. Therefore, the concept of “Dilated cardiomyopathy: A disease of the intercalated disc” [145] could be extended to a much broader spectrum of heart diseases potentially also including non-inflammatory diseases such as aortic stenosis. Once cellular contacts between cardiomyocytes are restored, anti-inflammatory mediators shut down the infiltration of OSM-releasing macrophages, allowing surviving cardiomyocytes to increase their contractile mass by responding to hypertrophic signals, thereby accelerating cardiac regeneration (Figure 6). Conversely, extended activation of the OSM signaling cascade converts hypertrophic signals into increased fetal remodeling and degeneration of the oxygen-consuming machinery, ultimately leading to reduced cardiac output and heart failure (Figure 6).

## 11. Summary and Conclusions

Oncostatin M is the most potent cytokine identified to date as an inducer of fetal adult cardiomyocyte remodeling. Its ability to activate two distinct receptor complexes—the type I LIFR/gp130 and the type II OSMR/gp130 receptor complexes—makes this cytokine the most pleiotropic member of the IL-6 family of cytokines in humans. Markers for OSM-activated signaling cascades can be detected in the acutely and chronically injured myocardium as well as under inflammatory and non-inflammatory disease conditions. Activation of OSM signaling cascades is protective under acute stress conditions but chronically contributes to the development of heart failure. The ability of OSM to integrate signals from unrelated receptors accelerates and potentiates fetal remodeling of adult cardiomyocytes. Moreover, as a potent inductor of extracellular protein expression and secretion, OSM exerts potent effects on the cardiac environment and potentially also on the systemic environment. Given the imbalance in research studies between interleukin-6 and the other members of the IL-6 cytokine family, it is time to reconsider the involvement of the OSMR/gp130 as well as the LIFR/gp130 complexes in human cardiovascular diseases.

## Figures and Tables

**Figure 1 ijms-23-01811-f001:**
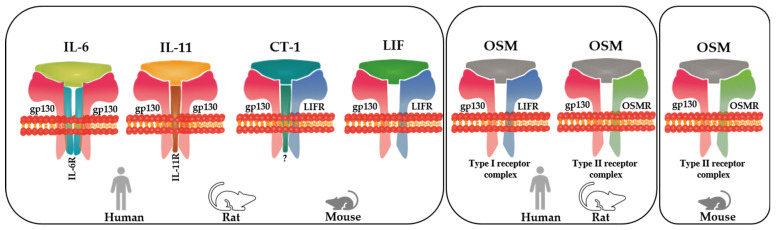
Simplified scheme illustrating the members of the interleukin-6 family of cytokines and their receptor complexes involved in adult cardiomyocyte remodeling and the differential formation of receptor complexes by OSM in the human, rat, and mouse. IL-6, IL-11, CT-1, LIF, and OSM elicit receptor complex activation in cardiomyocytes. Amongst them, OSM showed the strongest morphological effect on cultured cardiomyocytes, whereas the other members exerted a strong and comparable effect, except for IL-6, which was less effective because the weakly expressed IL-6 receptor probably needs trans-signaling events. The common co-receptor gp130 may explain the similar morphological responses of cardiomyocytes exposed to the IL-6 family. IL-6 and IL-11 activate two homodimeric gp130 receptor complexes containing the IL-6 receptor and the IL-11 receptor, respectively. In contrast, CT-1, LIF, and OSM do not require an additional non-signaling receptor to form functional complexes because they signal through a complex consisting of gp130 and the LIFR. OSM is unique in terms of receptor binding in that it can bind to both the type I (LIFR/gp130) receptor complex and type II (OSMR/gp130) receptor complex in humans and rats, whereas it binds only to the type I complex in mice.

**Figure 2 ijms-23-01811-f002:**
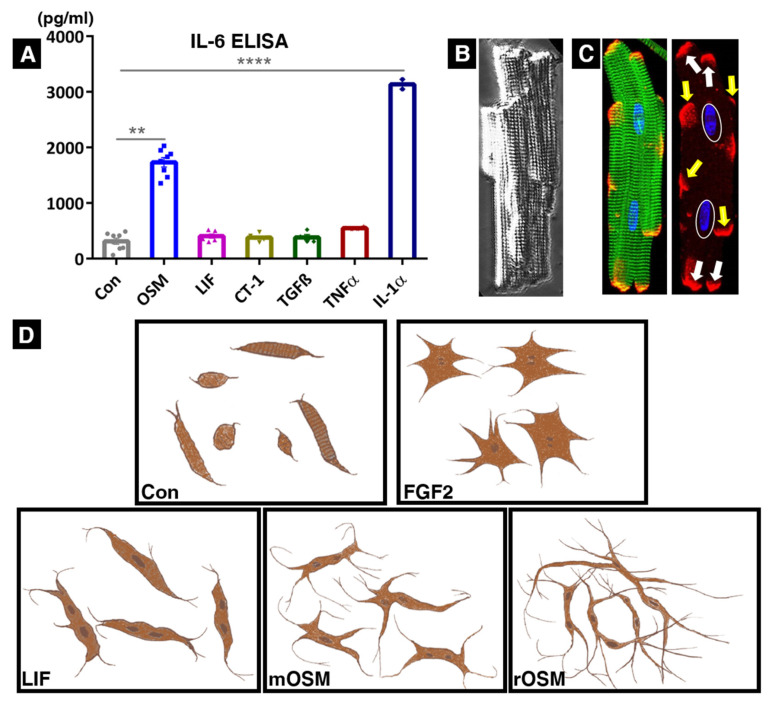
Schematic representation of morphological effects of activated type I and type II receptor complexes on cultured adult rat cardiomyocytes and IL-6 secretion. In all our screens, we utilized 20 ng/mL of albumin (Con), rat leukemia inhibitory factor (LIF, activates the rat LIFR/gp130 complex), mouse oncostatin M (mOSM, activates the rat OSMR/gp130 receptor complex), rat oncostatin M (rOSM, activates the rat LIFR/gp130 and the rat OSMR/gp130 complex), cardiotrophin-1 (CT1), transforming growth factor-β (TGFβ), tumor necrosis factor-α (TNFα), interleukin-1α (IL-1α), and fibroblast growth factor-2 (FGF2). (**A**) IL-6 ELISA showing concentrations of IL-6 in the cardiomyocyte culture supernatants 36 h after stimulation with albumin (*n* = 8), OSM (*n* = 8), LIF (*n* = 6), CT-1 (*n* = 6), TGFß (*n* = 4), TNFα (*n* = 4), and IL-1α (*n* = 2). Data represent the mean ± SEM. Statistical analysis was performed through an unpaired t test with Welch’s correction showing significances between Con and OSM **** *p* < 0.0001 and between Con and IL-1α ** *p* < 0.002. (**B**) Bright-field micrograph of a freshly isolated rat cardiomyocyte showing a complex three-dimensional structure with typical cross-striation. (**C**) Intercalated discs (stained with connexin-43 in red) are organized laterally (yellow arrows) and at the cell ends (white arrows). White ovals indicate nuclei and the green color marks sarcomeres (stained with sarcomeric α-actinin) in a freshly isolated cardiomyocyte. (**D**) Scheme summarizing the main morphological effects of rOSM, mOSM, LIF, and FGF2 after 5–7 days in culture. Cellular elongation and formation of multiple extensions were the most obvious changes after oncostatin M treatment. While elongation after rOSM stimulation was dominant, a certain amount of spreading might have become visible with increased culture time when cellular contacts were reestablished. It is important to note that serum (fetal calf serum, FCS) was only utilized for the initial plating of cardiomyocytes and then the cultures were kept without serum or any further growth enhancer/stimulant in these studies. For comparison, FGF-2-stimulated cultures show an increase in surface area but comparatively little elongation or formation of extensions.

**Figure 3 ijms-23-01811-f003:**
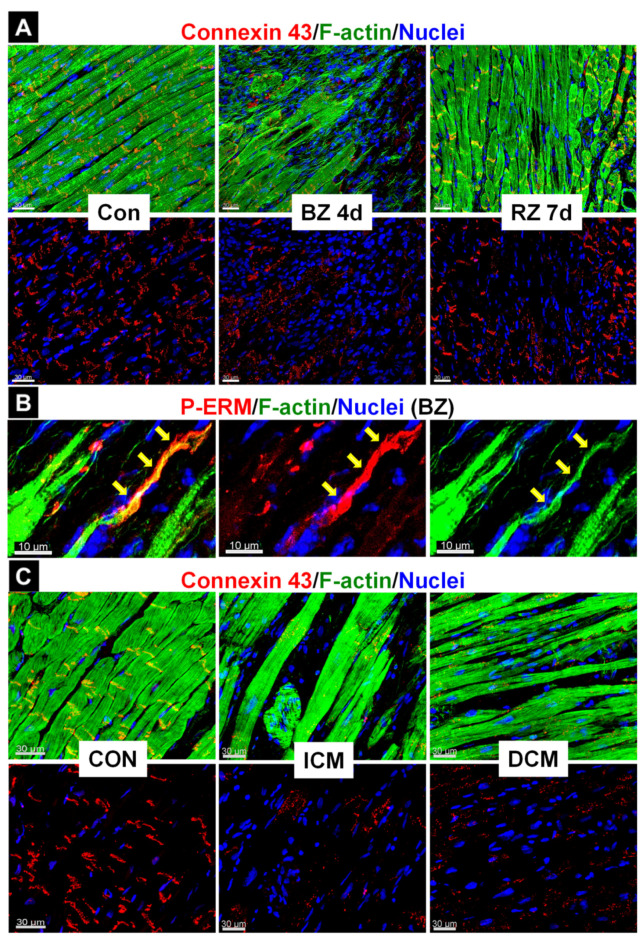
The architecture of intercalated discs is changed in the diseased myocardium. (**A**) Connexin 43, which marks intercalated discs in the normal heart, is downregulated/rearranged in cardiomyocytes of the border and remote zone in the infarcted mouse myocardium. (**B**) P-ERM (a marker of activated Ezrin/Radixin/Moesin proteins) characterizes the formation of cardiomyocyte extensions (yellow arrows) in the infarcted mouse myocardium. (**C**) Similarly, connexin 43 is downregulated in patients with ischemic (ICM) and dilated (DCM) cardiomyopathy, indicating remodeling of intercalated discs. CON represents human control tissue.

**Figure 4 ijms-23-01811-f004:**
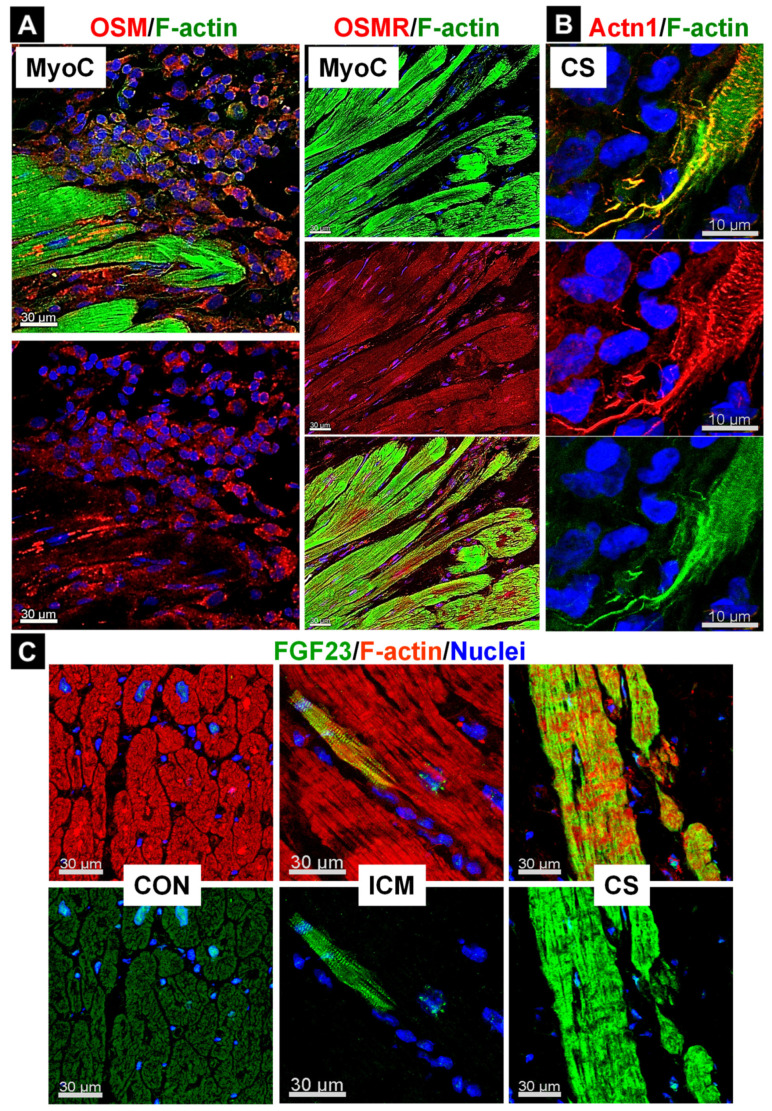
Infiltration of OSM-releasing cells is associated with fetal remodeling, increased expression of the OSMR, and secreted proteins in the human myocardium. MyoC indicates myocarditis, CS is cardiac sarcoidosis, ICM means ischemic cardiomyopathy, and CON represents human control tissue. Actn1 is non-muscle α-actinin-1 (Actn1). (**A**) Increases in OSM-positive infiltrates (mainly macrophages) are consistent with an increased expression of the OSMR in cardiomyocytes. (**B**) Increased dedifferentiation characterizes fetal remodeling (image is modified from [13]). Note the strong and thin cardiomyocyte elongation. (**C**) Both single FGF23-positive and clusters of positive cardiomyocytes can be identified. OSM is currently the only known cytokine that induces FGF23 expression in cardiomyocytes.

**Figure 5 ijms-23-01811-f005:**
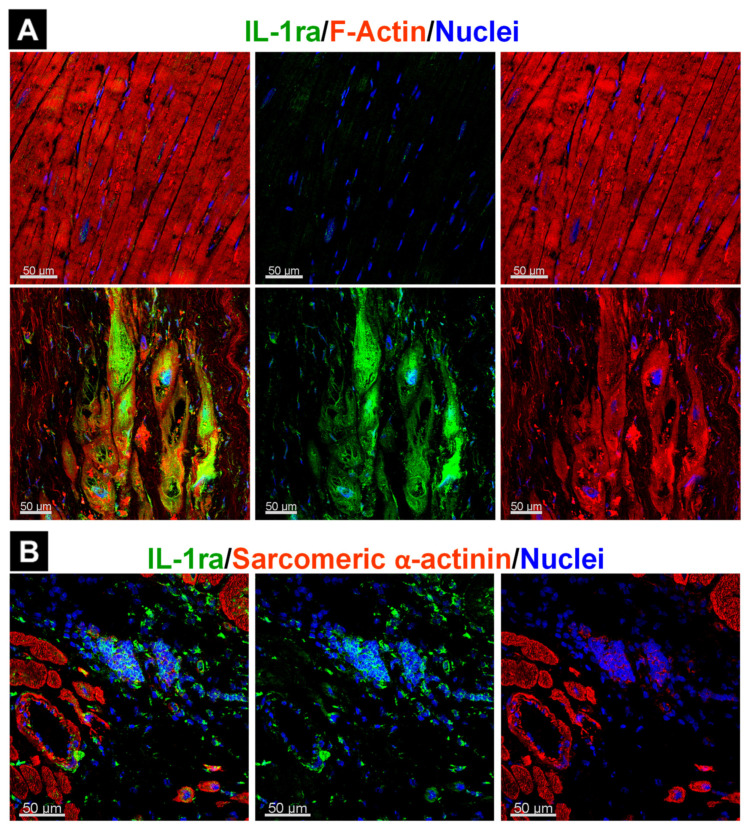
Localization of the interleukin-1 receptor antagonist (IL-1ra) is influenced by the underlying disease. (**A**) IL-1ra in cardiomyocytes of a patient with ischemic cardiomyopathy. (**B**) IL-1ra in the granuloma of a patient with cardiac sarcoidosis.

**Figure 6 ijms-23-01811-f006:**
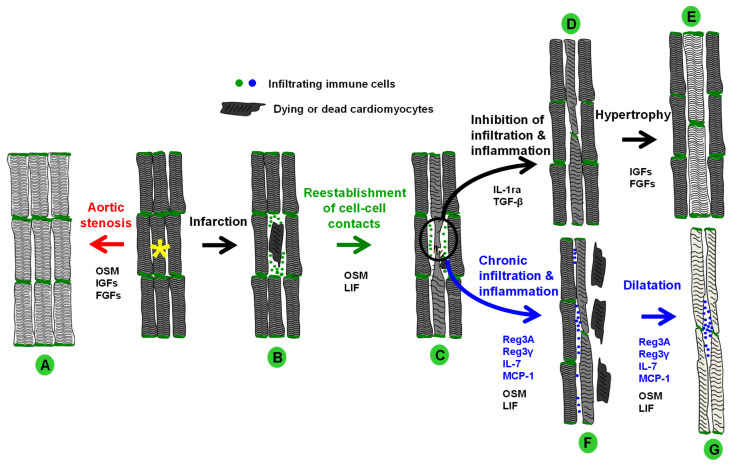
Hypothetical overall model of OSM-driven protection, regeneration, and failure of the heart. (**A**) The development of hypertrophy is initiated by activation of the OSMR in patients with aortic stenosis. When activation of the OSMR cascades decreases, the amount of hypertrophic signals increases (FGFs, IGFs). (**B**) After a cardiac injury such as acute myocardial infarction, OSM-releasing infiltrates reduce damage/infarct expansion and extension by inducing fetal remodeling of cardiomyocytes. Macrophage infiltration is controlled by various chemokine families (Reg3, IL-7, MCPs). (**C**) Cardiomyocytes form extensions and restore cell-cell contacts. (**D**) Infiltration and inflammatory processes are downregulated by anti-inflammatory molecules (IL-1ra, TGF-β) and hypertrophic pathways are activated (FGFs, IGFs). (**E**) Fetal remodeling is downregulated and myocytes undergo hypertrophic remodeling. Surviving cardiomyocytes adapt to the increased workload by enlarging and accumulating sarcomeres. (**F**) If infiltration and inflammatory processes persist, chronically activated OSM receptors cause degeneration of cardiomyocytes. (**G**) Dying cells and elongation of surviving cardiomyocytes lead to dilatation of the myocardium.

**Table 1 ijms-23-01811-t001:** Characteristic effects of activated oncostatin M type I (LIFR/gp130) and type II (OSMR/gp130) receptor complexes in cultured adult rat cardiomyocytes. The relative magnitude of effect is indicated by slight or no change (↑↓) and the increasing number of small arrows demonstrates the increasing or decreasing strength of cardiomyocyte responses. With an extended culture time, there might be a slight increase (↑↓/↑). Not determined is abbreviated as n.d. All effects can also be observed in mouse models of myocardial diseases and human diseased hearts as discussed in the text. Except for c-Kit, which was detected only by RT-PCR (due to the lack of a functioning anti-rat antibody), all regulated genes were observed on the mRNA as well as at the protein level. Explanation is provided in the text.

Treatment		Albumin	rOSM	mOSM	rLIF
Activated Receptors		None	OSMR/gp130 & LIFR/gp130	OSMR/gp130	LIFR/gp130
**Morphological Effects**					
Elongation		**↑↓**	↑↑↑	↑↑	↑
Increases in diameter		**↑↓**	**↑↓**	**↑↓**	↑
Increases in surface area		**↑↓**	**↑↓**	**↑↓**	↑
Beating activity		**↑↓**	**↓↓↓**	↓↓	n.d.
**Regulated Genes**	**Gene Symbol**				
**(Reexpressed Genes)**					
Smooth muscle α-actin	Acta2	**↑↓**	↑↑↑	↑↑	↑
Non-muscle α-actinin 1	Actn1	**↑↓**	↑↑↑	↑↑↑	n.d.
Non-muscle α-actinin 4	Actn4	**↑↓**	↑	↑	n.d.
Moesin	Msn	**↑↓**	↑↑↑	↑↑	↑
Runx1	Runx1	**↑↓**	↑↑↑	↑↑	**↑↓**
c-Kit	Kit	**↑↓**	n.d.	↑↑	**↑↓**
**Sarcomeric Loss**					
Sarcomeric α-actinin	Actn2	**↑↓**	↓	↓	↑↓
Myomesin1/2	Myom1/2	**↑↓**	↓↓↓	↓↓	↓
Myosin heavy chain 6, cardiac	Myh6 (α-MHC)	**↑↓**	↓↓↓	↓↓	↓
Myosin heavy chain 7, cardiac	Myh7 (β-MHC)	**↑↓**	↓↓↓	↓	↓
Myoglobin (O_2_ carrier)	Mb	**↑↓**	↓↓↓	↓↓	↓
**Secreted**					
Reg3β (& Reg3γ)	Reg3b (& Reg3g)	**↑↓**	↑↑↑ (n.d.)	↑↑	↑↑ (n.d.)
ANP	Nppa	**↑↓**	n.d.	↑↑	n.d.
BNP	Nppb	**↑↓**	n.d.	↑↑	n.d.
FGF23	FGF23	**↑↓**	↑↑↑↑↑	↑↑↑	↑↓/↑
Interleukin-6	IL6	**↑↓**	↑↑↑	↑↑↑	**↑↓**
TIMP-1	TIMP1	**↑↓**	↑↑↑	↑↑↑	**↑↓**/↑

## Data Availability

The data presented in this study are available in the article.

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
