# Peer review of "The Role of Oncostatin M and Its Receptor Complexes in Cardiomyocyte Protection, Regeneration, and Failure"

_ijms, 2022, doi:10.3390/ijms23031811_

Round 1
Reviewer 1 Report
Please see the attachment.

Author Response
This review by Kubin et al. discusses the importance of cytokines, more specifically interleukins in acute and chronic heart disease. The study highlights very well that heart failure should be viewed as a complex disease where immune-modulatory factors are involved instead of a simple consequence of hemodynamic and neurohormonal mechanisms. Another important message is that IL-6 family members should be researched more thoroughly in the future, as they were neglected before, while IL-6 has been studied extensively.
The focus of the work is an IL-6 family member, oncostatin M, which seems to be the most pleiotropic cardioactive cytokine of the family. The article summarizes experiments that revealed the properties of this molecule in sufficient detail. However, the focus around oncostatin M is not always evident, and the relevance of other mentioned molecules and discoveries is not clear in every case. There are some observations that can be considered regarding the text.
Major comments
- In the introduction, some background information on cytokines might be provided.
We agree with the reviewer and added few sentences.
- Generally, key phenomens and molecules should be introduced briefly before discussing them, e.g., fetal remodeling, FGF23, hypertrophic growth factors, TGF-ß to make their relevance clearer.
We would like to emphasize that fetal remodeling and dedifferentiation are the same and has been briefly introduced in section 5. We have added introductory sentences about other mentioned molecules.
- A short description of gp130 coreceptor would be reasonable, with stronger emphasis on its expression in all human cells.
We thank the reviewer to highlight it. We have given more information about gp130 in relevant sections.
- Classical IL-6 signaling and trans-signaling is presented and compared adequately, but it should be pointed out clearly how these mechanisms are similar to the signaling of oncostatin M in cardiac tissue. In fact, separately presenting oncostatin M receptor binding and downstream signaling in detail might be considered.
We are glad that we could explain IL-6 signaling adequately, but comparing the signaling between IL-6 and OSM is not the main aim of this review. Although both of them activate same pathways, the morphological changes that occur vary immensely. Although it has been shown that IL-6 induces hypertrophy in cardiomyocytes [ https://pubmed.ncbi.nlm.nih.gov/15717324/ ], but in our hands we did not observe such marked increase of surface area by IL-6. In fact, IL-6 showed the least morphological effects in comparison with IL-11, CT-1, LIF, and OSM. We did not do any gene expression analysis after IL-6 stimulation, so we can’t say that there might be overlapping expression patterns between OSM and IL-6. Coming to the downstream signaling effects of OSM, we have listed them in table 1.
- It is not evident whether the presented signaling mechanisms are exclusive to cardiac tissue.
As this review was about cardiac aspects, we did not talk about signaling in other cells. IL-6 family cytokines can activate Jak/STAT, MEK/ERK, ERK5, SAPK/JNK, p38, and PI3 kinase/Akt pathways in cardiomyocytes. Some signaling downstream targets like Timp1 and Reg family members are also expressed in smooth muscle cells and fibroblasts upon OSM stimulation. As it was irrelevant in this review, we did not point out the differences in signaling activation among all the cells expressing OSMR. For example, Erk5 pathway is activated in cardiomyocytes, but not in smooth muscle cells.
- It should be emphasized strongly when first mentioned, that the ability of oncostatin M to bind two types of receptor complexes is presumably responsible for its great pleiotropy in cardiac tissue. The two receptor complexes should be categorized to “type I” and “type II” in the text, as only the description of Figure 2., Table 1. and the conclusion declare clearly which complex is which type.
We appreciate the reviewer’s suggestion. We have elaborated this in the section describing morphological changes of cardiomyocyte cultures. We followed the suggestion of the reviewer and changed the text to type I and type II in text where ever it was relevant to differentiate the complexes.
- Signaling pathways activated by the IL-6 family members are listed, but basics and consequences of the JAK/STAT and MAPK pathways could be described somewhere in the text, as they are mentioned later but their significance is not explained.
We now have introduced the list of pathways activated by OSM in cardiomyocytes and specified that the major pathway through which the OSM mediated dedifferentiation occurs is via MEK/Erk pathway. For this reason, we prioritized focusing on the consequence of MEK/Erk signaling, but not on dissecting the pathway. The main consequence of Jak/STAT pathway activation by OSM is the expression of Reg proteins, which leads to infiltration of macrophages [ https://doi.org/10.1038/nm.3816 ], but we preferred not to elaborate it.
- It is not evident why considering JAK/STAT activation part of the IL-6 cascade neglects other family members’ involvement in heart disease. (Row 149-150).
As an example of how IL-6 was in limelight when ever gp130 or STAT3 were discussed, we have included a reference where IL-6 was predominantly examined in mice mutant for gp130 and STAT3, but did not show if other IL-6 family members were differentially regulated in their microarray data. We are confident that OSM might also have been differentially regulated in those mutant mice, but it has not been prioritized. We have shown earlier that OSM is expressed in the infarct border zone [ https://doi.org/10.1016/j.stem.2011.08.013 ], which was again confirmed by another group [ https://doi.org/10.1093/eurheartj/ehab724.1366 ].
- Hypothesized roles of IL-6 in heart disease should be highlighted before presenting results of IL-6 studies, e.g., it is not explained why it is associated with evocation of cardiac phenotype or myocardial abnormalities. (Row 168-169)
We have retracted this statement as it was wrongly placed and redundant. Our intention was to highlight the lackluster results from mutant mice for IL-6 which might be due to compensatory activation of other IL-6 family members.
- It is not clear why TGFß is put in parallel with OSM. (Row 418-419)
This statement was made to tell the contrasting ways with which OSM and TGFß affect cardiomyocyte contractility. OSM reduces contractility by rearrangement of contractile elements and by reduction of myofilaments [ https://doi.org/10.1016/j.stem.2011.08.013 ], where as TGFß reduces contractility by an increase in reactive oxygen species and NADPH oxidase [ https://doi.org/10.1093/abbs/gmy007 ].
- It might be reasonable to list which cell types express OSMR and secrete OSM, and to explain the origin of OSM in cardiac tissue explicitly.
In myocardium, OSM is mostly secreted by macrophages and is not expressed by other cells, where as OSMR is expressed by cardiomyocytes, fibroblasts, smooth muscle cells, and endothelial cells. OSMR is also expressed in wide range of cells in many other organs. We did not elaborate on this aspect as this information is well known and then it has to be done for all IL-6 family members and their receptors.
- Although FGF23 is specific for OSM and is mentioned several times in the article, it is not introduced at all.
This issue has been already raised in the second question of the reviewer and it has been addressed.
- Possibly every indicator molecule of OSM activity should be summarized collectively before mentioning them.
We apologize for not being able to understand the question properly. But we assume that the reviewer would like to see all molecules regulated by OSM together. We have listed the molecules regulated by OSM in Table 1 and hopeful that this is sufficient.
- It is not clear whether the MAPK pathway mediates fetal remodeling, if so, it should be highlighted clearly. (Row 514)
We thank the reviewer for pointing it out, we now have stated in the text that MEK/Erk pathway mediates dedifferentiation of cardiomyocytes.
- Basic introduction on the intercalated disc should not be in the second last, overviewing section of the text. (Row 654) It might even be unnecessary in this advanced paper, especially since much less evident terms are not explained.
We appreciate the suggestion of the reviewer and we agree. We have removed it now.
Minor comments (All Praveen)
- The introduction does not mention cardiac sarcoidosis in the list of heart diseases, but it is discussed later several times as an important example of heart disease, where oncostatin M is involved.
It is now listed in the abstract and in the introduction of cardiac diseases.
- It is not explained what the question mark under CT-1 represents in figure 1., ant is pushes the focus to that molecule.
In 1997, Robledo et al. (https://doi.org/10.1074/jbc.272.8.4855.) proposed that CT-1 binds to a α-receptor subunit, but so far, the corresponding cDNA could not be cloned. As nobody disproved the existence of a specific receptors, it has been always indicated with a question mark in the literature.
- In row 145, calling the respective primary receptors co-receptors is confusing.
We have replaced “coreceptors” with “specific receptors”
- After presenting effects of OSM on atherosclerosis, cardiac effects could be discussed in a new paragraph to make the features more categorized. (Row 270)
We agree with the reviewer and separated the atherosclerosis aspect into a separate paragraph.
- When listing features of OSM effects with ordinal numbers, “and” between every point is confusing, implying the last point every time. (Row 554-560)
It is true and misleading to use “and”. We have replaced it with a semi colon.
- Row 581: Abbreviation “DIRA” would be more expressive after the full term than inserted in it.
We have added the full form of DIRA
- It would be logical to state at the beginning of the paragraph starting at row 589 that cardiac sarcoidosis is the subject of the presented studies.
This paragraph was not specific for cardiac sarcoidosis, but about OSM mediated Reg3α and Reg3γ expression in ischemic heart disease and cardiac sarcoidosis.
- In figure 6., it is not obvious how the illustrated events follow from each other and that these factors ultimately lead to heart failure. OSM involvement is not labelled on the aortic stenosis branch of the figure.
The last few sentences of discussion part explains the chain of events. The numbering and labeling used in the scheme were not optimal and we have rearranged the scheme. The numbers which were used for description in the figure legend might have caused the confusion. We now have rearranged the numbers under each stage of cardiac remodeling. OSM and hypertrophic signals are not initiated simultaneously, but as pointed by the reviewer OSM has been added to the aortic stenosis.
- IL-7 is only mentioned in the second last section and the description of figure 6., therefore, its relevance is not obvious.
The reason to mention IL-7 briefly was only to specify another marker of cardiac macrophages which express IL-7 receptor α (IL-7Rα). We have extensively studied earlier about cardiac macrophages expressing IL-7Rα (https://doi.org/10.1016/j.cyto.2020.155053)
- Typographical errors noticed: Fig. 2.: ‘Il1α” should be capitalized to IL-1α. Row 372-373: “cells” possibly missing from the term “microvascular endothelial cells”. Row 486: space missing between “receptor” and “and”. Row 594: term missing after “cardiac” (sarcoidosis?).
Again, we are thankful for pointing out the typos. We have corrected them.
Reviewer 2 Report
Although inflammation is a major pathophysiology process in the heart failure (HF) and interleukin (IL)-6 can be an important inflammatory mediator leading to HF, the physiological roles of other members of IL-6 family has been relatively neglected. Recent accumulation of knowledge has revealed that Oncostatin M (OSM), a member of the IL-6 family, may play versatile roles and contribute to cardiac remodeling and heart failure progression. This manuscript mainly focuses on OSM and overviews the OSM-induced cardiomyocyte remodeling.
The main text is intriguing, well-organized and informative, in which the preclinical findings of OSM in cardiomyocytes are comprehensively summarized. There are minor concerns in the figures and table, where reconsideration and additional elaboration will be appreciated by the readers joined in this research area recently.
Specific comments:
Abstract
In lines 30-32, the logical connection between these two sentences is a little ambiguous. Do the authors try to point out the discrepancy between the animal model and actual human patients? It would be great if this could be clarified.
Figure1
A bit more elaboration or redesigning for each item would be appreciated regarding:
a) why the OSMs are colored differently (gray and red)
b) the presentation looks like that the text “IL-6R” indicates the whole complex including gp130. Please clarify if only the cyan part indicates IL-6R,
c) the meaning of “strongest effects”
Figure 2
Panel A, the purpose of this panel is uncertain. Why was mouse OSM added to rat cardiomyocytes? Does this panel need to be an actual data? If so, a bit more detailed method should be provided, such as culture media, treatment time and the way of statistical analysis. Some p-values are missing.
Panel C, please elaborate what kind of staining are these.
Table 1
What does “Not determined” really mean? Why does it come with small arrows (in Reg3b and 3g)?
Figure 3&4
It’s confusing that the magnifications are varied among images. In Fig4C, the orientations of tissue section even look different between control and the other areas. Why aren’t unified?
Figure 5A
Typo ? F-Aktin
Figure 6
What do the blue, black and highlighted texts indicate respectively?
Please eliminate hyphen (-) from each item. This looks like “minus”, which is confusing.
Author Response
Abstract
In lines 30-32, the logical connection between these two sentences is a little ambiguous. Do the authors try to point out the discrepancy between the animal model and actual human patients? It would be great if this could be clarified.
The reviewer’s point is valid. The discrepancy is unrelated to disease but species-specific. We have rephrased the sentence.
Figure1
A bit more elaboration or redesigning for each item would be appreciated regarding:
a) why the OSMs are colored differently (gray and red)
The idea was to differentiate mouse OSM, but we agree with the reviewer that it is misleading. We have unified the color.
b) the presentation looks like that the text “IL-6R” indicates the whole complex including gp130. Please clarify if only the cyan part indicates IL-6R,
Again, we agree with the reviewer, IL-6R is only the cyan part. We have modified the text orientation in the figure for better understanding.
- c) the meaning of “strongest effects”
The intention was to tell that, OSM exerts the most drastic morphological effects on cardiomyocytes. To be more specific in addressing the strong effects, the term “morphological” has been added.
Figure 2
Panel A, the purpose of this panel is uncertain. Why was mouse OSM added to rat cardiomyocytes? Does this panel need to be an actual data? If so, a bit more detailed method should be provided, such as culture media, treatment time and the way of statistical analysis. Some p-values are missing.
This indeed needs to be addressed properly. In our experience, mouse OSM binds to rat cardiomyocytes with similar affinity as rat OSM, but as rat OSM also activates LIFR/gp130 complex we used mouse OSM to show that IL-6 expression is mediated only by OSMR/gp130 complex (which is also indirectly confirmed by lack of IL-6 expression upon treatment with LIF). It is actual data, and we have replaced the graph which is more plausible to understand and elaborated the details of culture and statistics in the figure legend.
Panel C, please elaborate what kind of staining are these.
The details of the staining have also been updated
Table 1
What does “Not determined” really mean? Why does it come with small arrows (in Reg3b and 3g)?
“Not determined” means it has not been tested. The small arrows were unintentional and it might have happened while converting the table to a jpeg picture. We have replaced the picture with the original table.
Figure 3&4
It’s confusing that the magnifications are varied among images. In Fig4C, the orientations of tissue section even look different between control and the other areas. Why aren’t unified?
We should have been more careful with the magnifications. The images have been replaced with the same magnifications. Coming to the orientation of tissue sections, it is well known that the orientation of cardiomyocytes varies in the heart (https://doi.org/10.1038/s41598-018-24622-6). Unfortunately, due to the limited availability of control samples, we couldn’t find the sections with similar orientations. We have also shown earlier that FGF23 is not expressed in control human hearts (https://www.longdom.org/open-access/oncostatin-m-induces-fgf23-expression-in-cardiomyocytes-48545.html).
Figure 5A
Typo ? F-Aktin
Yes, it is a typo and has been rectified.
Figure 6
What do the blue, black and highlighted texts indicate respectively?
The choice of using different colors is to differentiate between different phases of cardiac remodeling. For the phase after infarction, the text “Reestablishment of cell-cell contacts” is colored green along with green dots which represent the early influx of infiltrating cells.
In the later chronic phase, with continuously expressed chemoattractants (blue colored) more infiltrating cells are recruited which are shown as “blue dots”.
The numbering and labeling was not optimal, so we have rearranged the scheme for better understanding.
Please eliminate hyphen (-) from each item. This looks like “minus”, which is confusing.
We appreciate the keen observation of the reviewer; the hyphens have been removed.
Reviewer 3 Report
This work aimed to provide an overview of Oncostatin M-induced cardiomyocyte remodeling and discuss the consequences of OSMR/gp130 and LIFR/gp130 activation under acute and chronic conditions.
The issue of Oncostatin M and receptors on cardiomyocyte remodeling is important. This work is interesting and important. It is well organized, however, there are some points must clarify in the manuscript.
- The therapy of acute and chronic cardiovascular disease through Oncostatin M and receptor have to discuss in this manuscript.
- All the abbreviations must identify at the first time in the text.
- On page 11, the title of Table 1 must show on the top of the table rather than the bottom of it.
- There are some type errors in the manuscript. For example, Figure legends of Figure 6 and line 475 Interleukin-6. Please revise the manuscript thoroughly.
- Line 486, receptorand must be “receptor and”
Author Response
1. The therapy of acute and chronic cardiovascular disease through Oncostatin M and receptor have to discuss in this manuscript.
We would like to bring it to the notice of the reviewer that direct targeting of OSM signaling, as a therapy, in humans is currently unavailable. As mentioned in the review, we could improve the cardiac function in a mouse model of dilated cardiomyopathy after inhibition of OSM signaling by injecting antibodies against, receptor binding domain, of OSMR. Moreover, we have also mentioned that acute activation of OSM after myocardial infarction rescues the mice. Although we did not specify the exact therapeutic approach from a clinical perspective, but in the last few sentences of the discussion we tried to point out the stages of remodeling where therapeutic intervention is necessary to overcome adverse remodeling.
2. All the abbreviations must identify at the first time in the text.
We appreciate the reviewer for the suggestion. We have gone through and the abbreviations have been added at the first instance of mentioning.
3. On page 11, the title of Table 1 must show on the top of the table rather than the bottom of it.
We thank the reviewer to pointing out. The Table title has been moved at the top of the table
4. There are some type errors in the manuscript. For example, Figure legends of Figure 6 and line 475 Interleukin-6. Please revise the manuscript thoroughly.
We apologize for the errors. They have been rectified.
5. Line 486, receptorand must be “receptor and”
We thank you again the reviewer, it has been corrected.